# A Review of the Use of Hyperthermic Intraperitoneal Chemotherapy for Peritoneal Malignancy in Pediatric Patients

**DOI:** 10.3390/cancers15102815

**Published:** 2023-05-18

**Authors:** David J. Byrwa, Clare J. Twist, Joseph Skitzki, Elizabeth Repasky, P. Ben Ham, Ajay Gupta

**Affiliations:** 1Department of Pediatrics, University at Buffalo Jacobs School of Medicine and Biomedical Sciences, Buffalo, NY 14203, USA; 2Division of Pediatric Oncology, Roswell Park Comprehensive Cancer Center, Buffalo, NY 14203, USA; 3Department of Surgical Oncology, Roswell Park Comprehensive Cancer Center, Buffalo, NY 14203, USA; 4Department of Immunology, Roswell Park Comprehensive Cancer Center, Buffalo, NY 14203, USA; 5Department of Surgery, Division of Pediatric Surgery, John R Oishei Children’s Hospital, Buffalo, NY 14203, USA

**Keywords:** hyperthermic intraperitoneal chemotherapy, HIPEC, soft-tissue sarcoma, desmoplastic small-round-cell tumor, cytoreductive surgery, CRS, peritoneal carcinomatosis, peritoneal malignancy

## Abstract

**Simple Summary:**

Hyperthermic intraperitoneal chemotherapy (HIPEC) is used to target microscopic peritoneal disease which can remain after visible disease has been surgically removed. It is used in the management of multiple adult cancer types, yet its use in the pediatric population is limited. This review paper provides an overview of the use of this modality in pediatrics in order to identify chemotherapy choice, document reported post-operative morbidity and mortality, and evaluate impact on overall survival. The use of HIPEC, most commonly with cisplatin, is generally tolerable with short-term post-operative complications and no reported post-operative mortality, yet the impact on overall survival versus systemic chemotherapy and debulking surgery is uncertain due to lack of clinical trials and small sample size. Continued gathering of outcome data of pediatric patients treated with HIPEC will aid the rational and safe application of HIPEC to pediatric peritoneal malignancies.

**Abstract:**

Hyperthermic intraperitoneal chemotherapy (HIPEC) can directly target microscopic peritoneal disease, has achieved regular consideration in the treatment of several adult cancer types, and is more recently being studied in pediatrics. This review paper provides an overview of the use of this modality in pediatrics in order to identify medication choice, discuss post-operative morbidity and mortality, and evaluate impact on overall survival. Four databases were searched including Scopus, PubMed, Embase, and CINAHL and ultimately 37 papers documenting the use of this modality comprising 264 pediatric patients were included. Malignancies treated include desmoplastic small round cell tumor, rhabdomyosarcoma, angiosarcoma, colorectal carcinoma, and mesothelioma, with several rarer tumor types. Cisplatin was the most commonly used drug for HIPEC at varying concentrations for 30–90 min in duration at temperatures of approximately 41–42 °C. Reported toxicities were generally self-limited and there was no post-operative mortality. The impact on overall survival versus systemic chemotherapy and debulking surgery is uncertain due to lack of clinical trials and very small sample size across tumor subsets and the overall pediatric population. The relationship between degree of tumor burden and extent of surgical debulking needs to be further clarified. Future directions include prospective clinical trials, establishment of patient databases to facilitate standardization of HIPEC in pediatric patients, and additional approaches to optimize HIPEC.

## 1. Introduction

The peritoneal spread of malignancy presents a significant medical and surgical challenge. Traditional systemic chemotherapy, bulk surgical resection, and/or radiation therapy in combination may provide temporary benefit but often are unable to eradicate all residual macroscopic and microscopic disease. Hyperthermic intraperitoneal chemotherapy (HIPEC) is a surgical procedure which requires the infusion of a heated chemotherapeutic agent into the abdomen followed by agitation of the abdomen and subsequent evacuation. The combination of HIPEC with cytoreductive surgery (CRS), a particular surgical technique, directly targets peritoneal surfaces that traditional intravenous (IV) chemotherapy would have difficulty penetrating due to the peritoneal–plasma barrier. This allows for both increased dose-intensity to the local peritoneal surface and destruction of microscopic remnants of malignancy [1]. HIPEC has been added to the aforementioned combination regimens and has achieved regular consideration in the treatment of adult peritoneal carcinomatosis or sarcomatosis.

In the adult population, HIPEC has been studied and utilized specifically for certain ovarian and gastrointestinal, including colorectal, cancers and has been associated with improved median overall survival (OS) and longer recurrence-free survival in small randomized controlled trials, and did not result in higher rates of side effects [2,3]; indications for HIPEC in adults are an active area of ongoing research due to these findings and are reviewed elsewhere [4]. These studies provide the impetus for the application of HIPEC as a component of multi-modality care for the management of several different malignancies metastatic to the peritoneum in the pediatric population.

The data on HIPEC use in children with peritoneal malignancies is unfortunately limited to single-arm and single-institution non-randomized clinical trials, extended case series, and case reports; due to the limited patient numbers, there are no multi-center randomized controlled trials investigating the potential benefit of HIPEC in this population. In part, this situation exists due to the need for the procedure to be performed at centers with surgeons that have the requisite expertise. Moreover, the inclusion criteria for treatment are typically strict and selected operable patients need to have normal organ function and tumors that can be safely resected and without distant metastases, with the presence and number of treatable lung and liver metastases debated in adults [5]. Therefore, the reports of HIPEC in pediatric patients likely represent a highly selected group of patients. Nevertheless, for young patients with peritoneal spread of aggressive tumors, HIPEC may offer a novel treatment strategy in combination with multi-modal therapy.

We review here the different types of HIPEC used in pediatrics, reported toxicities and complications, and its application by pediatric tumor type with a focus on soft-tissue sarcomas, including desmoplastic small-round-cell tumor (DSRCT), mesothelioma, rhabdomyosarcoma (RMS), and angiosarcoma, but also colorectal carcinoma, ovarian tumors, and other rare tumors in children. We conclude with future directions in how HIPEC and post-operative management may be further optimized. The summary of these reports and historical data may facilitate the consideration of including HIPEC in the treatment of children with peritoneal spread of these malignancies with the ultimate hope of improving survival in prospective clinical trials.

## 2. Methods

Without limiting the publication date, we systematically searched Scopus, PubMed, Embase, and CINAHL in February 2023. The search strategy utilized in all databases was as follows: (“hyperthermic intraperitoneal chemotherapy” OR “hyperthermic intraperitoneal perfusion with chemotherapy” OR hipec) AND (pediatric OR adolescent). No filters were applied. Four databases were searched including Scopus, PubMed, Embase, and CINAHL. Manual searching was also performed in the references of relevant articles for additional studies. The Scopus search yielded 258 results, PubMed yielded 229 results, Embase yielded 68 results, and CINAHL yielded 18 results, for a total of 573 results. A total of 250 duplicated records were removed before screening. Reasons for records that were excluded after screening include studies without pediatric patients, incorrect intervention, or incorrect focus, such as discussing anesthetic complications. Review papers, abstracts and conference papers were excluded from the review. Language was not considered as part of inclusion criteria although studies were excluded if an English version could not be obtained. This review is not officially considered a systematic review as not all sections during literature review were completed as part of the PRISMA 2020 checklist [6].

Data that was extracted from each source include type of malignancy, type of HIPEC, tumor burden score measured via peritoneal cancer index, degree of cytoreduction, temperature and duration of HIPEC, as well as what other modalities were used (i.e., bulk surgery, systemic chemotherapy, or radiation therapy). Acute toxicity, post-operative surgical complications, and outcome status were also noted. The data were synthesized in a table and basic statistics including percentages, mean, and median were calculated using Microsoft Office (Microsoft, Redmond, WA, USA).

## 3. Results

Ultimately, 49 publications were sought after publications were excluded as specified above and 37 papers were included in the review. A flow chart describing the process of paper selection is shown in Figure 1 [6].

In total, 264 pediatric patients were included in this review. The most common malignancy was DSRCT (87), followed by mesothelioma (64), RMS (30), primary ovarian tumors (19), colon carcinoma (11), undifferentiated sarcoma (6), Wilms tumor (6), hepatocellular (5), and angiosarcoma (4), with the rest made up by other tumor types. Cisplatin was the most commonly used drug, either alone or in combination with other chemotherapy, for HIPEC at varying concentrations for 30–90 min in duration at temperatures of approximately 41–42 °C, yet there was slight variation present across these parameters; this data can be found in Table 1. Reported toxicities and complications were generally self-limited with the most common being pancytopenia, anemia, or thrombocytopenia; Figure 2 shows adverse effects. Post-operative mortality was considered within 30 days of procedure and there was no reported post-operative mortality. Tumor burden scoring measured via peritoneal cancer index, degree of cytoreduction, post-operative morbidity rate, and survival outcomes can be seen in Table 2 and Figure 3. Degree of tumor burden was not consistently reported. No evidence of disease was noted in 33% of subjects at last follow-up, although length of follow-up varied across the papers. Of patients surviving at time of last follow-up, median length of follow-up was 19.5 months with a standard deviation of 24.1 months.

## 4. Discussion

### 4.1. Cytoreductive Surgery and HIPEC Technique

Degree of tumor burden is typically measured by the peritoneal cancer index (PCI) which considers both location and tumor size [44]. Tissue thickness plays a role in chemotherapy permeability and may affect tumor response; therefore, minimal thickness of any residual disease may lead to better outcomes. Unfortunately, this was not included in many of the case reports, and this limits our ability to make sense of how this factor may be associated with outcome status. In addition, optimal completeness of cytoreduction (CCR) is not strictly defined; traditionally, this is given a score and defined as a CCR0 (all visible disease removed), CCR1 (residual tumor nodules 2.5 mm or less), CCR2 (residual tumor nodules 2.5 to 25 mm), and CCR3 (residual tumor nodules > 25 mm), although some studies have opted to regard a complete resection as tumor less than 10 or 25 mm in size [44]. Upon achieving CRS, the subsequent closed-technique HIPEC is characterized by the following procedure: insertion of several temperature probes throughout the peritoneal cavity in order to ensure equal distribution of perfusate and protection of the liver from excess hyperthermia, the placement of inflow and outflow catheters sewn in place and connected to a circuit containing perfusate, temporary closure of the abdomen, heating of the perfusate to inflow temperature of 42 °C, agitation of the abdomen to ensure equal distribution of perfusate, and addition of chosen chemotherapy to the perfusate for the requisite time, followed by evacuation of perfusate and irrigation of the abdomen. The closure of the abdomen during HIPEC allows for more stable intraoperative conditions [45], although there are novel techniques reported for situations with lower tumor burden such as a laparoscopic approach to the closed abdomen to improve distribution of heat and chemotherapy [46].

A summary of studies describing the use of HIPEC in pediatric cancer, including the chemotherapy agents utilized as well as procedural details such as timing and temperature, is listed in Table 1. In the majority of papers reviewed, cisplatin was used; its benefits include synergy with hyperthermia and less than 10% of intraperitoneal dose systemic absorption, which allows for a high direct dose to the intraperitoneal tumor cells without systemic absorption and toxicity [47]. Alternative intraperitoneal chemotherapy agents include mitomycin C, oxaliplatin, doxorubicin, carboplatin, and paclitaxel [48].

### 4.2. Mitigating Agent and Reported Toxicities

Sodium thiosulfate (STS) has been used with HIPEC in both adult and pediatric series to reduce risk of toxicity [11,14,18]. The efficacy of STS in mitigating the risk of cisplatin-induced ototoxicity in children was studied in a randomized phase III trial through the Children’s Oncology Group (COG) trial ACCL0431 which enrolled pediatric patients with newly diagnosed solid tumors treated with systemic cisplatin-containing chemotherapy regimens. This study demonstrated that STS significantly reduced the risk of cisplatin-induced hearing loss; however, patients with metastatic disease who received STS had a significantly decreased event-free and overall survival compared to those who did not receive STS [49]. This difference was not seen in the patients with localized disease, raising the concern that, for high-risk patients with metastatic disease, STS may influence the effectiveness of the chemotherapy [49]. A series of 17 adult patients with intraperitoneal tumors treated with cisplatin HIPEC plus or minus STS found that higher doses of cisplatin (as high as 270 mg/m^2^) could be used without increased serum creatinine or myelosuppression when concurrent STS was used [50]. More recently, adult patients receiving cisplatin HIPEC plus or minus STS were shown to have significantly less renal impairment with the use of STS (0% versus 31.4%) [51]. In an expert panel, the use of STS with HIPEC was labeled as a weak positive recommendation with consensus [52]. Adequate pre- and post-hydration is typically also used for nephroprotection and may be most critical at the time of HIPEC as systemic vasodilation occurs due to hyperthermia which may impair renal perfusion.

### 4.3. Known Toxicities and Complications

The toxicities of HIPEC have been described in large studies of adults and have included, in one retrospective German series of over 2000 patients, a 10% risk of enteric fistulas and anastomotic leaks, 15% risk of reoperation, and 2.3% overall 30-day postoperative hospital mortality [53]. Moreover, this study commented on rate and grade of adverse surgical events utilizing the Clavien–Dindo system and reported almost 20% of patients suffered grade III (requiring surgical, endoscopic, or radiological intervention) or grade IV (life-threatening complications requiring ICU management) Clavien–Dindo complications. In a randomized controlled trial of 105 adult patients with colorectal cancer peritoneal carcinomatosis, HIPEC resulted in an 8% mortality rate and 35% morbidity, rates higher than other referenced studies, likely due to early experience and less stringent patient selection in these study cohorts [3]. Promisingly, a more contemporary study over 11 years of 325 adult patients receiving first-time HIPEC with median follow-up of 24 months identified low rates of long-term complications, with 18% of those receiving a one-time treatment of CRS with HIPEC incurring major late complications, including small bowel obstruction (5%), fistulas (3%), ureteral obstruction (2%), major vascular thrombosis (1%), and impaired gastrointestinal absorption (1%) [54]. In another study prospectively following the use of HIPEC in 356 adults with appendiceal mucinous malignancy, the total 30-day mortality or in-hospital mortality rate was 2.0%, while overall grade IV morbidity was 19% [55]. Overall, these data in adults can reasonably set expectations for what might occur in older pediatric patients; younger patients who are likely to have fewer co-morbidities may possibly tolerate the procedure with fewer complications.

The toxicity data that does exist in pediatrics is limited but overall is consistent with that described in adults receiving HIPEC. A summary of reported toxicity, surgical complications, and overall post-operative morbidity and mortality from published reports of pediatric HIPEC therapy are summarized in Figure 2. No post-operative mortality was reported in any of the studies referenced. One group reported their experience with 27 HIPEC procedures involving 23 patients aged 3 to 21 years; the dose-limiting toxicity for HIPEC with cisplatin was renal failure in three patients as well as two grade 3 hematologic toxicities, two grade 3 hepatic toxicities, and one grade 3 ileus with grading per the National Cancer Institute Common Toxicity Criteria v4.03 [12]. In a subsequent Phase II study by the same group, among 20 patients (including 14 with DSRCT) treated with HIPEC, there were two grade 4 creatinine elevations documented that spontaneously resolved within two weeks along with 16 grade 1 through 3 hematologic toxicities. Major postoperative complications occurred in 40% of the patients, including temporary neurogenic bladder, urinary tract infection, wound infection, abscess, and enterocutaneous fistula [14]. Of note, the patient who developed the enterocutaneous fistula had previously received unspecified high-dose abdominal radiation therapy. In a report of seven pediatric patients ages 12 to 18 years with peritoneal mesothelioma who received CRS and HIPEC with cisplatin, immediate post-operative events included pancreatitis, acute tubular necrosis, hyperbilirubinemia, bilateral pleural effusions, pneumothorax, and two cases each of anemia and coagulopathy, all resolving with supportive care [19]. In a French retrospective series of 22 patients, 90% of whom were treated with an open HIPEC technique, 64% of patients had complications within 30 days of HIPEC, including grade 4 toxicities of hemoperitoneum, gastric fistulas, urinary fistula, bilious peritonitis, pulmonary embolism (1), and aponeurectomy for compartment syndrome in the calf (1) and grade 3 toxicities of pleural effusion requiring drainage (3), septic ascites (1), urinary tract infection (1), and severe anorexia (1) [25].

A pediatric study of DSRCT reported the long-term complications of CRS, HIPEC, and whole abdominal radiotherapy (WART), with the most impactful complications including gastroparesis, adhesive bowel obstruction secondary to sclerosing peritonitis, and hemorrhagic cystitis [22]. The chronic inflammatory complication of sclerosing encapsulating peritonitis was also noted in a case study of a 13-year-old patient 13 months after receiving CRS, HIPEC, and WART as part of treatment for DSRCT [21]. It has been postulated that the possibility of morbidity is cumulative among systemic chemotherapy, CRS/HIPEC, and WART and, thus, comparable oncologic outcome with multimodality use would inherently have to be weighed against toxicity [22]. This idea has been shown in the adult population, in that those who received iterative CRS/HIPEC versus those who received singular CRS/HIPEC had higher rates of fistula and overall major late complications [54]. As a result, the inclusion of WART with HIPEC should be approached with caution.

### 4.4. Pediatric Applications of HIPEC

#### 4.4.1. Desmoplastic Small-Round-Cell Tumor (DSRCT)

DSRCT is an aggressive sarcoma of adolescents and young adults that primarily arises in the abdomen and has a predilection for peritoneal spread, with a 5-year OS of 18.1% in a recent SEER analysis [56]. HIPEC may be considered in the management of these patients based on mixed evidence from mostly retrospective case series over the past 30 years. A retrospective review compared the outcome of 24 DSRCT patients (ages 5 to 43) from 1995 to 2008 treated as follows: patients who received chemotherapy alone without surgery, patients who received chemotherapy plus debulking surgery of at least 90% of tumor, and patients who received chemotherapy, HIPEC, and CRS (10 mm or less) [9]. In comparing those that received CRS with HIPEC versus debulking surgery, the 3-year median survivals were 71% and 62%, respectively, and the result was not statistically significant. However, the overall survival in the group who received chemotherapy alone was only 26%, which was statistically significant. The study also confirmed the survival benefit of having no extra-abdominal metastases. The same group conducted a single-arm, single-institution phase 2 trial from 2012 to 2013 in which 20 patients ages 23 months to 50 years, 14 of whom had DSRCT with median PCI 15, received CRS (25 mm or less) and HIPEC [14]. Patients with resectable liver metastases were allowed on study and all patients had to demonstrate neoadjuvant chemotherapy-responsive disease of partial response or better. In addition, all DSRCT patients received WART. In this highly selected population, the 3-year OS for DSRCT patients was 79%, and the trial also demonstrated the survival benefit of not having distant metastases. However, 40% of patients had major postoperative complications, as described. The same group conducted a third study of 50 pediatric patients aged 3 to 21, the majority of whom had DSRCT [10]. CRS and HIPEC were associated with improved median OS, although this was dependent on extent of cytoreduction (25 mm or less) as well as PCI lower than 16.

In a smaller retrospective review, CRS and HIPEC with adjuvant WART was performed for nine DSRCT patients ages 10 to 24 years and resulted in a 3-year OS of 55% [22], with complications as previously described. A single-institution retrospective study of 187 DSRCT patients with a median age of 22.6 years found that the addition of HIPEC to CRS (25 mm or less) in neoadjuvant chemotherapy-responsive patients did not alter 3- or 5-year OS, although it did confirm the survival benefit of CRS alone [57]. Another retrospective multicenter study in France from 1991 to 2018 studied 100 patients with DSRCT with median age 25 and median follow-up of 103 months [58]. They identified predictive factors for cure in the five patients (5%) that survived and found PCI less than 12, disease stage, absence of extraperitoneal metastases, completeness of CRS (2.5 mm or less), and post-operative WART as positive predictive factors. HIPEC was administered in 15 of 71 patients who achieved complete cytoreduction and was not predictive of cure.

Aside from the above referenced studies with larger study sizes, there are multiple case reports described. In Australia, two patients with DSRCT aged 14 and 21 with PCI of 5 and 12 underwent CRS with HIPEC and were reported to be alive with disease 20 and 25 months after initial diagnosis [31]. Recently, the first pediatric patient to receive HIPEC in the United Kingdom was reported for a 7-year-old with DSRCT. It was tolerated well without complications and consolidated with WART and maintenance therapy with complete remission 18 months after CRS [26]. There is also the first reported case of pediatric DSRCT in Turkey, in which a 10-year-old received HIPEC with autologous hematopoietic stem cell transplant; however, this patient died 8 months later [30]. There is one case report of an 8-year-old with DSRCT with hepatic and peritoneal metastatic disease who received multi-modality therapy in the form of CRS with HIPEC, WART, and autologous hematopoietic cell transplant. This patient experienced relatively minimal side effects and was noted to be disease-free at 6 years [29]. The addition of autologous hematopoietic cell transplantation to CRS with HIPEC was also studied in a series of three pediatric patients with DSRCT with PCI ranging from 22 to 27 who had a median overall survival of 37.5 months with combined modalities (Siddiqui et al. 2020). Overall, DSRCT is the most studied malignancy in pediatrics for the use of HIPEC and the regimens were generally well-tolerated. It appears that it is most likely to be successful in patients that have the disease limited intra-abdominally, have an initial response to neoadjuvant chemotherapy, and have a successful cytoreductive surgery performed by centers experienced in the procedure. The use of WART and other consolidative therapies requires additional study.

#### 4.4.2. Rhabdomyosarcoma (RMS)

RMS is the most common soft-tissue sarcoma of childhood and patients with high-risk disease have a poor prognosis; rarely, RMS will metastasize to the peritoneum, providing opportunity for use of CRS with HIPEC in these select situations. A case series from China of seven RMS patients, ages 2 to 14 years, treated with CRS and doxorubicin and ifosfamide or cisplatin HIPEC as consolidation of up-front therapy (or first relapse in one patient) with PCI between 2 and 12 demonstrated no evidence of disease (NED) in six of the seven patients, the seventh alive with disease, with an average of 16 months of follow-up [27]. A German group studied CRS (2.5 mm or less) and HIPEC with cisplatin plus or minus doxorubicin in a group of six children between 1 and 5 years old with advanced or recurrent intra-abdominal RMS [8]. All patients showed no evidence of disease after a median 12 months of follow-up and the procedure was tolerated very well, with no grade 3 or 4 adverse events and two events of self-limited proteinuria. Patients were excluded from this study if they had extra-abdominal tumors or unresectable intra-abdominal lesions. A second case report details a 2-year-old with primary peritoneal fusion-negative rhabdomyosarcoma who was found to be neoadjuvant chemotherapy-responsive with a PCI of 8, and the decision was made to perform CRS followed by HIPEC if complete CRS was possible [24]. Cisplatin HIPEC was delivered after successful CRS, was uneventful with no complications, and the patient completed maintenance therapy with no evidence of disease at 10 months. The patient was able to avoid radiation therapy as a result of this approach. Lastly, there is a third case report which focuses on anesthetic concerns of CRS with cisplatin HIPEC in a two-year-old child. They reveal no post-operative morbidity or mortality; however, there was no mention of survival or length of disease-free state [28].

#### 4.4.3. Angiosarcoma

Angiosarcoma is a vascular malignancy that is rare in children and also rarely metastasizes to the peritoneal cavity but has been successfully treated in a few cases with HIPEC. An 11-year-old diagnosed with metastatic ovarian angiosarcoma with peritoneal sarcomatosis and malignant ascites was found to be in relapse after exploratory laparotomy and drainage of the ascites [15]. The patient was initially treated with ifosfamide and doxorubicin then switched to gemcitabine and docetaxel with partial response, further consolidated with CRS (20 mm or less) with oophorectomy and paclitaxel HIPEC, followed by adjuvant chemotherapy. The patient had no evidence of disease at 43 months after initial diagnosis but suffered ovarian failure secondary to the removal of both ovaries, for which she was placed on hormone replacement therapy. A case report of two additional pediatric patients with metastatic intra-abdominal angiosarcoma to the peritoneum with ascites further demonstrated the safety of an approach utilizing HIPEC [17]. In the first case, a 13-year-old presented with PCI of 16 and neoadjuvant chemotherapy-responsive disease and proceeded to CRS and HIPEC plus ultrasound ablation of hepatic capsule implants, followed by adjuvant trametinib and chemotherapy for an oncogenic NRAS mutation on tumor sequencing. There were no complications post-operatively. Unfortunately, the patient relapsed 10 months post-operatively and received salvage chemotherapy. In the second case, a 10-year-old presented with PCI of 17 and neoadjuvant chemotherapy-responsive disease and proceeded to CRS and HIPEC plus WART and adjuvant chemotherapy. There were no complications post-operatively. The patient was disease-free for a year, but then developed two liver metastases that were treated with radiofrequency ablation and remained disease-free. Lastly, there is a case report of a 13-year-old with angiosarcoma who underwent neoadjuvant chemotherapy followed by CRS with mitomycin C HIPEC [41]. This patient was noted to be disease-free at 12 months after diagnosis.

#### 4.4.4. Colorectal Carcinoma

In adults, the ESMO 2016 consensus guidelines for metastatic colorectal cancer advised that patients with PCI less than 12 and no evidence of systemic disease could consider HIPEC with complete CRS at experienced centers [59]. However, the PRODIGE 7 trial in 2018 found that HIPEC with oxaliplatin for 30 min after complete CRS did not increase OS compared to CRS alone and patients treated with HIPEC had higher 60-day morbidity [60]. Currently, consequently, the use of HIPEC in this population is debated with some favoring a longer-duration mitomycin-C-based treatment [61].

Treatment of pediatric peritoneal carcinomatosis from colorectal cancer was systematically reviewed in 2020 from five different sources by Sorrentino et al. [62]. PCI index was not evaluated. They included nine cases (one case treated twice for a total of ten HIPEC procedures) treated at five centers, ranging between 11 and 16 years of age, six of which had complete CRS (0 mm of tumor), three unknown, and one patient which did not receive any type of CRS. The most common subtype was signet cell carcinoma and all patients who had lymphadenectomies were found to have positive lymph nodes. All patients were treated with neoadjuvant chemotherapy, most commonly with the FOLFOX or FOLFIRI backbone therapy. The HIPEC was mitomycin C in six patients and cisplatin in three. The patient that was treated twice had a second HIPEC after mitomycin C with oxaliplatin. Three patients were free of disease with an average follow-up time of 74 weeks, and another three patients had local recurrences, two of whom died (one of which was the patient without CRS); there was no data on three other patients. The procedures were well-tolerated in all cases.

#### 4.4.5. Primary Disseminated Ovarian Tumors

While HIPEC is commonly used in adult patients with advanced epithelial ovarian cancer with support in the literature in a multicenter phase 3 trial [2], the evidence is more limited in pediatrics. In a group of eight pediatric patients aged 4 to 18 years with ovarian primary tumors and multifocal peritoneal disease limited to the abdomen, CRS (25 mm or less) followed by HIPEC with cisplatin demonstrated a median OS of 63% (three of the eight patients died) with an average of 25 months of follow-up, with a post-operative complication rate of 25%, including a patient with prior WART who developed an enterocutaneous fistula [13]. Two of the three patients who died had less than 20 mm of unresectable tumor left behind during CRS, and all three patients were heavily pre-treated and on their third and fourth salvage regimens. The histologic types of ovarian tumors in this report included three yolk-sac tumors, one Sertoli–Leydig cell tumor, one ovarian PNET, one choriocarcinoma, one juvenile granulosa cell tumor, and one adenocarcinoma. In a separate case report, an 11-year-old patient with Sertoli–Leydig cell tumor (SLCT) was treated with neoadjuvant chemotherapy followed by complete CRS and cisplatin HIPEC, but had an extraperitoneal recurrence two months after surgery and was alive with disease at 38 months of follow-up [18]. As part of an adult case series, one 19-year-old with SLCT received complete CRS with cisplatin HIPEC twice as part of her initial treatment and recurrence 7 months later and was alive without disease or significant post-operations complications 21 months from the original surgery [63].

#### 4.4.6. Mesothelioma

All of the studies evaluating the use of CRS with HIPEC in pediatric mesothelioma have been retrospective in nature, with the largest study compiled by The European Cooperative Study Group for Pediatric Rare Tumors which included 27 patients less than 18 years of age with malignant mesothelioma. Notably, 19 of these patients received CRS with HIPEC and median follow up was 6.7 years with 5-year overall survival of 82.3% [33]. Type of HIPEC used was not homogenous as six different combinations were utilized and there was no reporting of post-operative morbidity or mortality. The next-largest case series included a subset of nine pediatric patients who received CRS with HIPEC as part of multi-modality therapy, and follow-up ranged from 1.8 to 15 years although PCI was not consistently reported and type of HIPEC varied [38].

A retrospective cohort of pediatric patients with disseminated intraabdominal malignancies treated with CRS and HIPEC included two patients with mesothelioma [18]. They were treated with CRS (2.5 mm or less) and cisplatin HIPEC, followed by post-operative paclitaxel, with one patient receiving additional pemetrexed [18]. Neither patient suffered a recurrence and both had no evidence of disease at 20 and 48 months of follow-up. Separately, a retrospective study comprised seven patients ages 12 to 18 years with malignant peritoneal mesothelioma previously treated with surgery and/or chemotherapy [19]. Four had complete CRS and the other three had incomplete CRS, and there were no post-operative mortalities with toxicities as previously described. One patient had a repeat CRS and HIPEC for recurrence 120 months after initial CRS, two additional patients had repeat CRS for recurrence 48 and 80 months after initial CRS, and a fourth patient had a recurrence 4 months after initial CRS and went onto a clinical trial. Despite this poor event-free survival from first HIPEC, five of the seven patients (71%) were alive at last follow-up with median follow-up of about 10 years. A third retrospective study from France identified seven patients with peritoneal mesotheliomas treated with CRS and HIPEC (various agents including mitomycin, cisplatin, oxaliplatin, irinotecan, and doxorubicin) and three relapsed, but all seven were alive with a median follow-up of 60 months at time of publication [25].

#### 4.4.7. Wilms Tumor

There are at least three reported cases of HIPEC used successfully in patients with Wilms tumor. A 5-year-old with recurrent Wilms tumor was treated with neoadjuvant chemotherapy followed by complete CRS and cisplatin HIPEC, and tolerated them well, but had an extraperitoneal relapse 3 months after surgery and was alive with disease at 8 months of follow-up [18]. In another case report, a 12-year-old with Wilms tumor relapsed 6 months after the end of therapy with multiple intraabdominal tumor implants. Complete CRS and cisplatin HIPEC were utilized in this patient and there was no abdominal recurrence 12 months after surgery, although a left pleural recurrence required radiation treatment [11]. The regimen was well-tolerated, despite the single kidney, and the only complication was a wound infection which healed. In a third case report, a 22-month-old with Wilms tumor with a second recurrence with a PCI of 4 was treated with complete CRS, doxorubicin, and ifosfamide HIPEC with no complications and was disease-free 31 months from surgery [27].

#### 4.4.8. Undifferentiated Sarcoma

There is a case report of a 5-year-old with a localized 8.4 cm undifferentiated sarcoma of the prostate treated with doxorubicin, ifosfamide, and focal radiation with a partial response followed by preoperative proton beam therapy and delayed primary excision [23]. The complete CRS was consolidated with cisplatin HIPEC and sodium thiosulfate nephroprotection. Complications included ileus and enterococcus UTI. The patient was disease-free after 15 months. A second case report of a 7-year-old in second recurrence and PCI of 5 was treated with CRS and HIPEC with ifosfamide and doxorubicin but recurred 2 months after treatment and died of disease 10 months after surgery [27].

#### 4.4.9. Melanoma

Melanoma with metastatic spread to the peritoneal cavity is exceedingly rare and there is only one known case report in the literature involving children. In this report, a 3-year-old with congenital melanocytic nevus syndrome was discovered to have leptomeningeal melanoma and extensive abdominal disease. Initial therapy with radiation therapy, temozolomide, and sorafenib demonstrated near-complete response of leptomeningeal disease but continued abdominal disease with PCI of 12. CRS and cisplatin HIPEC were used and she remained free of disease seven months after treatment but later died of progressive leptomeningeal disease at an unknown time [11].

#### 4.4.10. Inflammatory Myofibroblastic Tumor (IMT)

There is one case report that utilized normothermic HIPEC (NIPEC) with CRS in the management of this rare tumor in a 5-month-old patient. The patient had an initial CRS procedure at presentation which revealed an ALK1-positive IMT. The decision was made after two weeks to pursue complete CRS and infuse doxorubicin NIPEC followed by adjuvant chemotherapy and crizotinib. At 12 months follow-up since treatment completion, the patient remains in complete remission [43].

#### 4.4.11. Epithelioid Inflammatory Myofibroblastic Sarcoma (EIMS)

A case report documented the use of ifosfamide and doxorubicin in a 5-year-old patient with a first relapse with EIMS and PCI of 2 treated with CRS and doxorubicin plus ifosfamide HIPEC who had no evidence of disease 5 months after treatment [27].

#### 4.4.12. Histiocytic Sarcoma

A case report of a 4-year-old patient with widespread intra-abdominal disease described the use of CRS with cisplatin HIPEC at 4.5 months post induction chemotherapy when disease progression was noted [39]. Post-operative complications were not described. Patient was disease-free at 7 years after diagnosis after completing multi-modality therapy.

#### 4.4.13. Others

A case series of patients from China with various rare tumors that metastasized to the peritoneum included single cases of children with malignant rhabdoid tumor of the kidney, immature teratoma, neuroblastoma, malignant germ cell tumor, nephroblastoma, and clear cell sarcoma of the kidney [27]. Median PCI for these tumor types ranged from 2–9, so disease was generally not extensive. This group had follow-up ranging from 3.5 to 31 months, with five deaths due to tumor recurrence. A French retrospective case series included three patients. The first was a 6-year-old with Ewing sarcoma with partial response to neoadjuvant chemotherapy and mitomycin plus cisplatin who received partial CRS and HIPEC plus WART and who was alive with disease with 1 year of follow-up. The second is a 15-year-old with fibrolamellar hepatocellular carcinoma with a partial response to neoadjuvant chemotherapy who underwent complete CRS plus oxaliplatin and irinotecan HIPEC who was alive with disease with 54 months of follow-up. The last is a 15-year-old with a solid pseudopapillary tumor of the pancreas who had complete CRS and up-front oxaliplatin plus irinotecan HIPEC who was alive with no evidence of disease 25 months after surgery [25].

Overall, the use of HIPEC, most commonly with cisplatin, is generally tolerated with short-term post-operative complications, yet the impact on overall survival versus systemic chemotherapy and debulking surgery is uncertain due to lack of clinical trials and small sample size for multiple tumor types. Moreover, some of the papers also included patients who underwent radiotherapy and stem cell transplant as part of their care, which may influence morbidity and survival as well. Of note, there was no post-operative mortality reported in any of the referenced pediatric literature.

## 5. Future Directions

While the use of CRS with HIPEC may potentially improve survival in patients with certain types of peritoneal sarcomatosis or carcinomatosis, the prognosis is still poor and further investigation into the optimization of HIPEC is required. Further optimization of HIPEC parameters have been reviewed elsewhere; it was determined that the optimal temperature for HIPEC is in the range of 40 to 43 degrees Celsius and that cisplatin and oxaliplatin demonstrate significantly better tumor kill after 60 min [64,65], which is in line with most HIPEC conducted in pediatrics.

One group has developed an animal model of HIPEC for alveolar RMS where they are able to measure an adapted PCI, alter HIPEC conditions and timing, and measure tumor apoptosis, although their ethics committee did not allow them to conduct CRS prior to HIPEC, limiting the utility of the model [66]. They followed up these initial studies with the evaluation of photodynamic therapy and the photosensitizer hypericin, an extract from St. John’s wort, in addition to cisplatin HIPEC in the same animal mouse model and demonstrated additional reduction of tumor proliferation [67]. It is additionally useful in its ability to allow better identification of the tumor tissue it is killing, down to tumors less than 1 mm, and will be a promising system to bring to humans after additional study. Other groups have studied peritoneal metastases-derived organoids to identify patient tumor sensitivity to mitomycin C, oxaliplatin, the combination, and other novel compounds, which could theoretically be used to tailor regimens in the future [68].

It may be important to use these models with clinical data to better understand how hyperthermia affects the body and enhances the antitumor response of the chemotherapy chosen and if that can be further tested and optimized. There is evidence that heat shock proteins (HSP) play an important role in cellular stress during HIPEC treatment, and specifically that HSP27 is upregulated with a rise in temperature. Therefore, paclitaxel could upregulate the Bax/Bcl-2 ratio by inhibiting HSP27 and promote apoptosis [69]. The sequence in which carboplatin, hyperthermia, and etoposide are given has shown to be either synergistic or antagonistic in vitro based on timing [70]. In mice injected with carcinoma cells and then either subjected to mild systemic heating or not, the mice that received heating demonstrated significantly enhanced tumor response compared to control mice. The heating appeared to alter the tumor microenvironment, interstitial fluid pressure, hypoxia, and perfusion [71]. Theoretically, these findings could justify a form of radiotherapy after HIPEC, perhaps less aggressive than WART given the concerns of toxicities when the two are given together. Patient tissues of peritoneal carcinomatosis were analyzed and found that optimal anti-tumor effects could be achieved by preselecting specific target temperatures to overcome highly conserved HSP mechanisms within tumor cells [72]. Additional studies have shown that nutritional stresses can induce the heat-shock promoter HSP70B, which could theoretically raise the threshold needed to produce an anti-tumor effect [73].

As a result of the aforementioned implications of nutritional stress on treatment as well as the poor oral intake experienced by many patients after undergoing CRS with HIPEC, it is worthwhile to consider novel ways to improve patient nutritional status in the peri-operative period. Immunonutrition is a combination of nutritional supplements including proteins, long-chained triglycerides, and amino acids such as arginine and glutamine meant to minimize post-surgical immunosuppression and improve visceral microperfusion [74]. A randomized controlled trial investigated the benefits of immunonutrition compared to standard nutritional feeds before and after CRS and HIPEC for 62 adult patients with peritoneal metastases and found trends but no statistical significance in length of hospitalization, wound infections, or postoperative complications, but may deserve further study [75]. Similar study in the pediatric population is warranted as a means to optimize outcomes to minimize morbidity while maximizing the treatment response to CRS with HIPEC.

While the majority of studies published in the literature were either retrospective study designs or case studies, prospective designs in the form of open trials for pediatric patients with intraperitoneal malignancy will be more informative. It is encouraging to note that some clinical trials based in adult centers for HIPEC had dropped the eligibility criteria to include patients in the pediatric age range, such as 16 or 17 years old for eligible patients with ovarian cancer (https://clinicaltrials.gov/ct2/show/NCT01144442 (accessed on 30 March 2023) and https://clinicaltrials.gov/ct2/show/NCT05246020 (accessed on 30 March 2023)). There is currently an open phase I trial utilizing HIPEC in children and young adults ages 1 to 25 years with resectable, refractory/recurrent abdominal or pelvic tumors who will undergo CRS followed by HIPEC with doxorubicin and cisplatin plus STS (https://clinicaltrials.gov/ct2/show/NCT04213794 (accessed on 30 March 2023)). Prospective studies such as these ones will be critical to informing clinical practice in the coming years.

Retrospectively and prospectively registering patients in an international pediatric HIPEC database would be extremely useful to help guide future endeavors in this field. While one hospital in Belgium is running such a registry (https://clinicaltrials.gov/ct2/show/NCT01617382 (accessed on 30 March 2023)), it is unclear how many patients are being registered internationally on the registry. The Peritoneal Surface Oncology Group International (PSOGI) also has a registry based on cases from adult surgical oncologists, focused on appendiceal neoplasms, mesothelioma, and rare tumors. This registry may be a good basis upon which to base or mirror pediatric efforts, which certainly deserve their own analysis. As our review has demonstrated, there is currently not standardization across centers with regard to what is considered acceptable cytoreduction, what are optimal HIPEC parameters, what chemotherapies should be used, and more. These types of registries can provide data to help answer other questions that are currently unknown in the pediatric patient population. For example, in adults, more than half of women with oophorectomy as part of CRS for intraperitoneal metastatic spread of colorectal or appendiceal origin had a microscopic synchronous ovarian metastases [76]—it is unknown what is the rate in pediatrics and if this rate of micrometastatic disease matters given that utmost attention is paid to attempting to preserve future fertility in our youngest patients.

## 6. Conclusions

This review brings together data related to reported toxicities, morbidity, mortality, and overall survival for a variety of pediatric peritoneal malignancies for which the utilization of CRS with HIPEC may be considered. One of the main limitations is related to the rarity of these tumors and the required specialization of cancer centers for these complex surgeries and treatments. Meticulous gathering of long-term outcome data of all previous pediatric patients treated with HIPEC and all patients moving forward in addition to prospective, multi-institutional randomized controlled trials will aid the rational and safe application of HIPEC to pediatric malignancies in hopes of yielding improvements in overall survival.

## Figures and Tables

**Figure 1 cancers-15-02815-f001:**
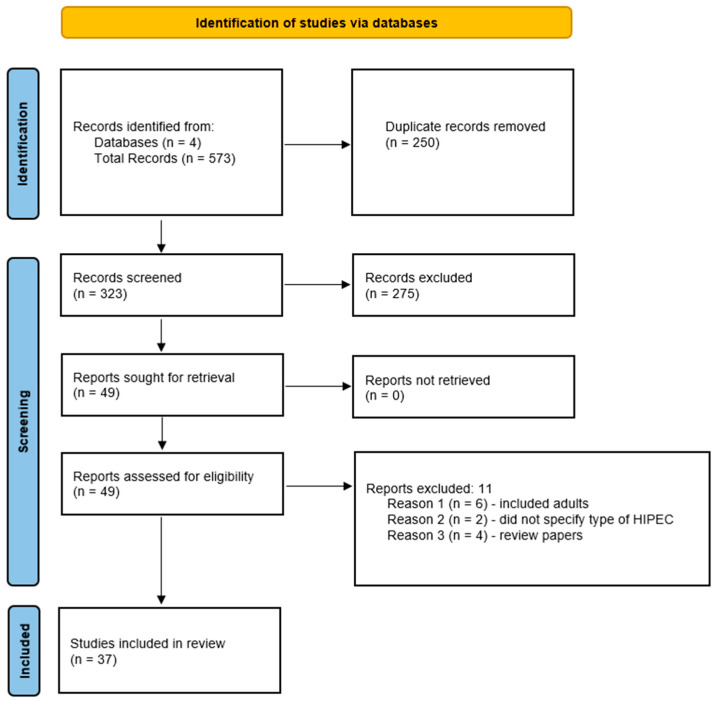
Flow chart describing the process of paper selection.

**Figure 2 cancers-15-02815-f002:**
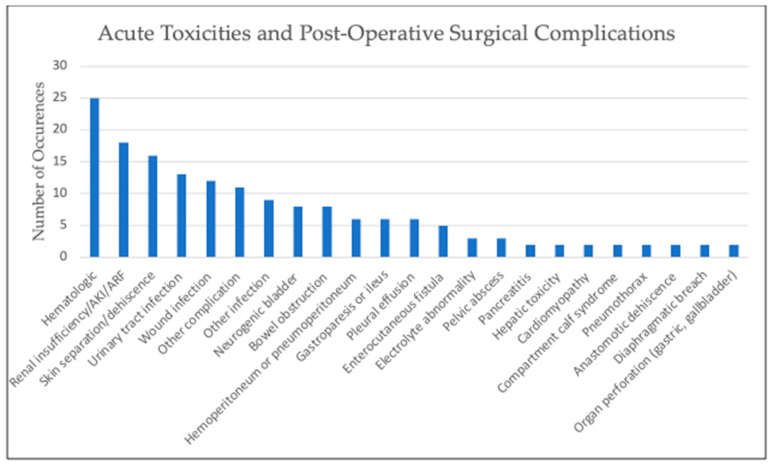
Graph of reported toxicities and surgical complications. “Hematologic” includes pancytopenia, anemia, or thrombocytopenia. “Other complication” includes one-time occurrences such as abdominal hematoma, cardiotoxicity, pancreatic leak, anorexia, elevated amylase levels, subclinical decrease in hearing, hyperbilirubinemia, transient femoral neuropathy, hemorrhage from right psoas, pulmonary embolism, and urinary obstruction. “Other infection” includes one-time occurrences such as perirectal abscess, bilious peritonitis, central venous catheter infection, bacteremia, pneumonia, sepsis, abdominal abscess, empyema, and bacterial diarrhea.

**Figure 3 cancers-15-02815-f003:**
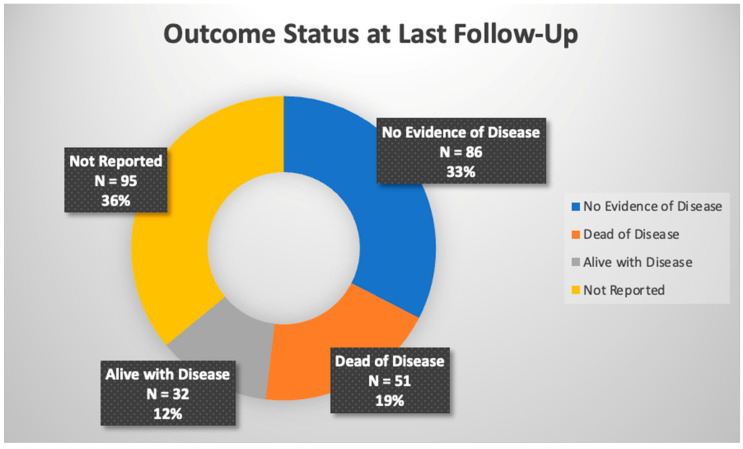
Overall outcome status across referenced studies.

**Table 1 cancers-15-02815-t001:** Type of Malignancy with Type, Temperature, and Duration of HIPEC.

Paper Referenced	Type of Malignancy	Type of HIPEC *	Temperature and Duration of HIPEC
Bautista et al. [7]	Ovarian tumors (3), mesothelioma (2), fibrolamellar hepatocellular carcinoma (2), other types (2)	Oxaliplatin 300 mg/m^2^ + Irinotecan 200 mg/m^2^	43 °C, 30 min
Gesche et al. [8]	RMS (6)	Cisplatin 37.5–75 mg/m^2^ (6) Doxorubicin 15 mg/m^2^ (4)	42.5 °C, 60 min
Hayes-Jordan et al. 2010 [9]	DSRCT (8)	Cisplatin 100–150 mg/m^2^ (8) + Mitoxantrone (1)	40–41 °C, 90 min
Hayes-Jordan et al. 2015 [10]	DSRCT (21), RMS (7), mesothelioma (4), other (18)	Cisplatin 100 mg/m^2^	41 °C, time unspecified
Hayes-Jordan et al. 2012 [11]	Melanoma (1), Wilms tumor (1)	Cisplatin 100 mg/m^2^	41 °C, 90 min
Hayes-Jordan et al. 2012 [12]	DSRCT (13), mesothelioma (5), RMS (2), Wilms tumor (2), other (5)	Cisplatin 150 mg/m^2^	40.5–41 °C, 90 min
Hayes-Jordan et al. 2016 [13]	Ovarian tumors (8)	Cisplatin 100 mg/m^2^	41 °C, 90 min
Hayes-Jordan et al. 2018 [14]	DSRCT (14), RMS (2), undifferentiated sarcoma (2), other (2)	Cisplatin 100 mg/m^2^	41°C, 90 min
Pariury et al. [15]	Angiosarcoma (1)	Paclitaxel 40 mg/m^2^	41.5 °C, 90 min
Sorrentino et al. [16]	Colon adenocarcinoma (1)	Mitomycin C	42 °C, 60 min
Winer et al. [17]	Angiosarcoma (2)	Mitomycin C 18 mg/m^2^ then 9 mg/m^2^ × 2 (1) Cisplatin 30 mg/m^2^ + Doxorubicin 100 mg/m^2^ (1)	41.5–42 °C 60–90 min
Zmora et al. [18]	RMS (3), mesothelioma (2), other (4)	Cisplatin 100 mg/m^2^ (7), Doxorubicin 20 mg/m^2^ (1), Mitomycin C (1)	41 °C, 90 min
Malekzadeh et al. [19]	Peritoneal mesothelioma (7)	Cisplatin 250 mg/m^2^ (7) + Paclitaxel 125 mg/m^2^ and 5-FU (3)	41.1–43 °C time unspecified
Reingruber et al. [20]	Colon adenocarcinoma (1)	Mitomycin C 30 mg/m^2^	41.2 °C, 90 min
Whitlock et al. [21]	DSRCT (1)	Cisplatin 100 mg/m^2^	40.0 °C, 90 min
Stiles et al. [22]	DSRCT (10)	Cisplatin 100 mg/m^2^ (8), Mitomycin-C (1), Melphalan (1)	42 °C, 60–90 min
Findlay et al. [23]	Undifferentiated sarcoma (1)	Cisplatin 100 mg/m^2^	42.5 °C, 90 min
Kazi et al. [24]	RMS (1)	Cisplatin 100 mg/m^2^	41.5–42°C, 90 min
Scalabre et al. [25]	Mesothelioma (7), DSRCT (7), RMS (1), ovarian tumors (2), other types (5)	Oxaliplatin 300 mg/m^2^ + Irinotecan 200 mg/m^2^ (7), Cisplatin + Doxorubicin (2), Oxaliplatin (2), Mitomycin + Cisplatin (9), Cisplatin + Ametycin (1), Cisplatin 200 mg/m^2^ (1)	41–44 °C, 30–90 min
Bexelius et al. [26]	DSRCT (1)	Cisplatin 100 mg/m^2^	41 °C, 90 min
Zhu et al. [27]	RMS (7), Wilms tumor (2), clear cell sarcoma of kidney (2), sarcoma (2), other types (6)	Doxorubicin 15 mg/m^2^ + Ifosfamide 1 g/m^2^ (11), Doxorubicin 15 mg/m^2^ + Cisplatin 50 mg/m^2^ (5), Cisplatin 50 mg/m^2^ (3)	40.5–41.5 °C, 60 min
Doctor et al. [28]	RMS (1)	Cisplatin	41.5–42.0°C, 60 min
Xiao et al. [29]	DSRCT (1)	Cisplatin	40.5 °C, 90 min
Kartal et al. [30]	DSRCT (1)	Irinotecan 200 mg/m^2^ + Oxaliplatin 300 mg/m^2^	41.0 °C, 60 min
Fan et al. [31]	DSRCT (2)	Cisplatin	41.5 °C, 90 min
Oyeniyi et al. [32]	Colon carcinoma (5)	Mitomycin C (3), Oxaliplatin (2)	NR
Orbach et al. [33]	Mesothelioma (27)	Cisplatin + Doxorubicin (9), Cisplatin + Mitomycin (5), Cisplatin (3), Oxaliplatin + Irinotecan (2), Cisplatin + Paclitaxel (1), unspecified (3)	NR
Siddiqui et al. [34]	DSRCT (3)	Cisplatin 100 mg/m^2^	Temperature unspecified, 90 min
Cacciotti et al. [35]	DSRCT (1)	Cisplatin	NR
Lamm et al. [36]	Mesothelioma (1)	Cisplatin	NR
Msika et al. [37]	DSRCT (3)	Cisplatin 75 mg/m^2^ + Mitomycin C	41.0–43.0°C, 30 min
Vermersch et al. [38]	Mesothelioma (9)	Cisplatin + Mitomycin (4), Oxaliplatin + Irinotecan (2), Cisplatin + Doxorubicin (2), Methotrexate (1)	42.0–43.0 °C, 30–60 min
Sandler et al. [39]	Histiocytic sarcoma (1)	Cisplatin	NR
Brecht et al. [40]	Mesothelioma (1)	Cisplatin	41.0°C, 60 min
El-Sharkawy et al. [41]	Angiosarcoma (1)	Mitomycin C	Temperature unspecified, 90 min
Vaz et al. [42]	Leiomyosarcoma (1)	Cisplatin + Doxorubicin	42.0°C, 60 min
Garnier et al. [43]	Inflammatory myofibroblastic tumor (1)	Doxorubicin	Temperature unspecified, 30 min

DSRCT: desmoplastic small-round-cell tumor; RMS: rhabdomyosarcoma; NR: not reported; min: minutes. * Concentration specified, otherwise not reported. Number included in parentheses is the number of patients who received that HIPEC if not all patients received the same HIPEC.

**Table 2 cancers-15-02815-t002:** Tumor Burden Score, Degree of Cytoreduction, Post-operative Morbidity, and Survival Outcomes.

Paper Referenced	Peritoneal Cancer Index	Degree of Cytoreduction	Morbidity Rate	Survival Status at Last Follow-Up (Median Follow-Up) *	NED
Bautista et al. [7]	NR	CCR0	7 out of 9	4 out of 9 (58.8 mos)	3 out of 9
Gesche et al. [8]	4–21	CCR0	2 out of 6	6 out of 6 (12 mos)	6 out of 6
Hayes-Jordan et al. 2010 [9]	3–33	CCR0	3 out of 8	5 out of 8 (36.3 mos)	1 out of 8
Hayes-Jordan et al. 2015 [10]	16 (reported median)	CCR0–CCR2	28 out of 50	Statistic not reported (21.9 mos)	NR
Hayes-Jordan et al. 2012 [11]	12	CCR0	1 out of 2	2 out of 2 (9.5 mos)	0 out of 2
Hayes-Jordan et al. 2012 [12]	NR	CCR0	16 out of 23	NR	26%
Hayes-Jordan et al. 2016 [13]	0–16	CCR0 (6), CCR1 (2)	2 out of 8	5 out of 8 (32 mos)	5 out of 8
Hayes-Jordan et al. 2018 [14]	15 (reported median)	CCR0–CCR1	NR	79% survival at 3 years in group with DSRCT	NR
Pariury et al. [15]	NR	NR	0 out of 1	1 out of 1 (43 mos)	1 out of 1
Sorrentino et al. [16]	3	CCR0	0 out of 1	1 out of 1 (11 mos)	1 out of 1
Winer et al. [17]	16–17	CCR0	NR	2 out of 2 (12 mos)	1 out of 2
Zmora et al. [18]	NR	CCR0	4 out of 9	7 out of 9 (28 mos)	4 out of 9
Malekzadeh et al. [19]	6–25	CCR0 (4), CCR1 (2), CCR2 (1)	4 out of 7	5 out of 7 (104 mos)	1 out of 7
Reingruber et al. [20]	NR	NR	1 out of 1	0 out of 1 (Death at 30 mos)	0 out of 1
Whitlock et al. [21]	NR	NR	0 out of 1	1 out of 1 (13 mos)	0 out of 1
Stiles et al. [22]	5–20	CCR0 (5), CCR1 (4), CCR2 (1)	9 out of 9	2 out of 9 (34 mos)	1 out of 9
Findlay et al. [23]	NR	CCR0	1 out of 1	1 out of 1 (14 mos)	1 out of 1
Kazi et al. [24]	8	NR	0 out of 1	1 out of 1 (10 mos)	1 out of 1
Scalabre et al. [25]	NR	CCR0 (16), CCR1 (4), CCR2 (2)	14 out of 22	13 out of 22 (57.5 mos)	6 out of 22
Bexelius et al. [26]	NR	NR	0 out of 1	1 out of 1 (26 mos)	1 out of 1
Zhu et al. [27]	2–21	CCR0 (13), CCR1 (6)	2 out of 19	14 out of 19 (14 mos)	13 out of 19
Doctor et al. [28]	6	NR	0 out of 1	NR	NR
Xiao et al. [29]	21	CCR0	1 out of 1	1 out of 1 (72 mos)	1 out of 1
Kartal et al. [30]	NR	NR	0 out of 1	0 out of 1 (8 mos)	0 out of 1
Fan et al. [31]	5, 12	NR	2 out of 2	2 out of 2 (22 mos)	0 out of 2
Oyeniyi et al. [32]	NR	CCR0	NR	3 out of 5 (10 mos)	3 out of 5
Orbach et al. [33]	NR	CCR0 11), CCR1 (4), CCR2 (2), remaining unspecified	NR	20 out of 27 (80.4 mos)	17 out of 27
Siddiqui et al. [34]	22–27	NR	0 out of 3	0 out of 3 (37.5 mos)	0 out of 3
Cacciotti et al. [35]	NR	NR	1 out of 1	0 out of 1 (38 mos)	0 out of 1
Lamm et al. [36]	NR	NR	1 out of 1	NR	NR
Msika et al. [37]	NR	CCR0 (2), CCR2 (1)	1 out of 3	1 out of 3 (10 mos)	1 out of 3
Vermersch et al. [38]	>10 for 4, unknown for others	CCR0 (5), CCR2 (3), unspecified (1)	3 out of 9	9 out of 9 (7 mos)	6 out of 9
Sandler et al. [39]	NR	NR	NR	1 out of 1 (72 mos)	1 out of 1
Brecht et al. [40]	NR	NR	0 out of 1	1 out of 1 (17 mos)	1 out of 1
El-Sharkawy et al. [41]	NR	NR	NR	1 out of 1 (12 mos)	1 out of 1
Vaz et al. [42]	NR	NR	0 out of 1	NR	NR
Garnier et al. [43]	NR	CCR0	NR	1 out of 1 (12 mos)	1 out of 1

CCR: Completeness of cytoreduction; NED: no evidence of disease; NR: not reported; mos: months. * Reported survival characteristics, either alive with disease or no evidence of death, are at time of publication.

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
