# Peer review of "A Review of the Use of Hyperthermic Intraperitoneal Chemotherapy for Peritoneal Malignancy in Pediatric Patients"

_cancers, 2023, doi:10.3390/cancers15102815_

Round 1
Reviewer 1 Report
In this work, the authors summarized the latest progress of the use of hyperthermic intraperitoneal chemotherapy in pediatrics in order to identify medication choice, discuss post-operative morbidity and mortality, and evaluate impact on overall survival. It is basically well organised and should be interest to the researcher in many related fields. It could be accepted for publication if the following concerns are properly addressed.
1. Although the focus on the output of this new technique, it is suggested the authors start with a short introduction of the basic theory of HIPEC and also briefly introduce some progress of HIPEC on the theoretical aspect.
2. The tables in this review are very long and it is hard to get important information from them. It is suggested to use other means to illustrate the results.
Author Response
- In this work, the authors summarized the latest progress of the use of hyperthermic intraperitoneal chemotherapy in pediatrics in order to identify medication choice, discuss post-operative morbidity and mortality, and evaluate impact on overall survival. It is basically well rganized and should be interest to the researcher in many related fields. It could be accepted for publication if the following concerns are properly addressed.
RESPONSE: Thank you for your positive comments, they are much appreciated.
- Although the focus on the output of this new technique, it is suggested the authors start with a short introduction of the basic theory of HIPEC and also briefly introduce some progress of HIPEC on the theoretical aspect.
RESPONSE: Thank you for your constructive comments. We moved some of our discussion points related to CRS/HIPEC theory to the introduction as well as added details related to progress of HIPEC which makes logical step for extrapolation of HIPEC use in children. Please see lines 50-65 in the paper, which are included below:
“Hyperthermic intraperitoneal chemotherapy (HIPEC) is a surgical procedure which requires the infusion of a heated chemotherapeutic agent into the abdomen followed by agitation of the abdomen and subsequent evacuation. The combination of HIPEC to cytoreductive surgery (CRS), a particular surgical technique, directly targets peritoneal surfaces that traditional intra-venous (IV) chemotherapy would have difficulty penetrating due to the peritoneal-plasma barrier. This allows for increased dose-intensity to both the local peritoneal surface and destruction of microscopic remnants of malignancy [6]. HIPEC has been added to the aforementioned combination regimens and has achieved regular consideration in the treatment of adult peritoneal carcinomatosis or sarcomatosis. In the adult population, HIPEC has been studied and utilized specifically for certain ovarian and gastrointestinal, including colorectal, cancers and has been associated with improved median overall survival (OS) and longer recurrence-free survival in small randomized controlled trials, and did not result in higher rates of side effects [1,2]; Indications for HIPEC in adults are an active area of ongoing research due to these findings and are reviewed elsewhere [3].”
- The tables in this review are very long and it is hard to get important information from them. It is suggested to use other means to illustrate the results.
RESPONSE: Thank you for your constructive suggestion. We removed toxicities and surgical complications from table 2 and converted this data to a graph which allows for better visualization (figure 2). We also removed mortality rate from table 2 because there was no meaningful data in that column of the table (no reported mortality across the studies). We moved peritoneal cancer index and completeness of cytoreduction data from table 1 to table 2. We also added figure 3 which shows overall outcome status at last follow up, which was not readily apparent before. Lastly, we believe switching the orientation on pages containing the tables to landscape allows for more cohesive visualization. Please see lines 136-145 which provides the most noteworthy results and helps direct readers to tables and figures. They are below here as well:
“Reported toxicities and complications were generally self-limited with the most common being medullary aplasia, anemia or thrombocytopenia; figure 2 shows adverse effects. Post-operative mortality was considered within 30 days of procedure and there was no reported post-operative mortality. Tumor burden scoring measured via peritoneal cancer index, degree of cytoreduction, post-operative morbidity rate and survival outcomes can be seen in table 2 and figure 3. Degree of tumor burden was not consistently reported. No evidence of disease was noted in 33% of subjects at last follow-up although length of follow-up varied across the papers. Of patients surviving at time of last follow-up, median length of follow-up was 19.5 months with a standard deviation of 24.1 months.”
Thank you again for taking the time to review our manuscript and provide constructive feedback so that we may revise and improve its quality. We hope the changes are sufficient for acceptance for publication in Cancers.
Reviewer 2 Report
Dear Editor,
The review paper provides an overview of the use of hyperthermic intraperitoneal chemotherapy This modality in pediatrics in order to identify medication choices, discuss post-operative morbid- and mortality, and evaluate the impact on overall survival
The review is descriptive more criticism needs to be added when the authors report the data
Moreover, a graphical representation of emerging finding on adverse effects recurrence and others could help the readers
add ref DOI: 10.3390/cancers14143414
Dear Editor,
The review paper provides an overview of the use of hyperthermic intraperitoneal chemotherapy This modality in pediatrics in order to identify medication choices, discuss post-operative morbid- and mortality, and evaluate the impact on overall survival
The review is descriptive more criticism needs to be added when the authors report the data
Moreover, a graphical representation of emerging finding on adverse effects recurrence and others could help the readers
please add the ref DOI: 10.3390/cancers14143414
Author Response
- The review paper provides an overview of the use of hyperthermic intraperitoneal chemotherapy This modality in pediatrics in order to identify medication choices, discuss post-operative morbid- and mortality, and evaluate the impact on overall survival. The review is descriptive more criticism needs to be added when the authors report the data.
RESPONSE: Thank you for your constructive comments. We have added below in our paper where there are summative statements for the data, which come from disparate populations, histotypes, techniques, etc. and are largely difficult to ascertain definitive conclusions from. Nevertheless, we have strived to outline where our paper is able to make a point at each juncture and, in the body of the text, have described each paper in-depth with regards to the limitations and successes of each attempt. The future directions and conclusions sections, not reproduced here, also attempt to summarize the current field, make critical comments on the work thus far, and point towards what might be helpful for future collaborations:
Lines 245-248
Overall, these data in adults can reasonably set expectations for what might occur in older pediatric patients; younger patients who are likely to have fewer co-morbidities may pos-sibly tolerate the procedure with fewer complications.
Lines 286-287
As a result, the inclusion of WART with HIPEC should be approached with caution.
Lines 344-348
Overall, DSRCT is the most studied malignancy in pediatrics for the use of HIPEC and the regimens were generally well-tolerated. It appears that it is most likely to be successful in patients that have disease limited intra-abdominally, initial response to neoadjuvant chemotherapy, associated with a successful cytoreduction, and performed by centers experienced in the procedure. The use of WART and other consolidative therapies requires additional study.
Lines 537-543
Overall, the use of HIPEC, most commonly with cisplatin, is generally tolerated with short-term post-operative complications yet the impact on overall survival versus system-ic chemotherapy and debulking surgery is uncertain due to lack of clinical trials and small sample size for multiple tumor types. Moreover, some of the papers also included patients who underwent radiotherapy and stem cell transplant as part of their care, which may influence survival as well. Of note, there was no post-operative mortality reported in any of the referenced pediatric literature.
- Moreover, a graphical representation of emerging finding on adverse effects recurrence and others could help the readers.
RESPONSE: Thank you for this suggestion. We removed adverse effects from table 2 and converted this data to a graph (figure 2) which allows for better visualization of the findings. We also created figure 3 to highlight outcome status of all subjects included in the review paper as this data is not readily available from table 2.
- Please add ref DOI: 3390/cancers14143414
RESPONSE: While we agree that the content discussed in this reference (“Sarcoma Common MHC-I Haplotype Restricts Tumor-Specific CD8+ T Cell Response”) has application to the future use of immunotherapy in management of sarcoma malignancy, this was not the focus of our paper; it is currently unknown how MHC haplotypes might affect HIPEC therapy or associated morbidity and mortality.
Thank you again for taking the time to review our manuscript and provide constructive feedback so that we may revise and improve its quality. We hope the changes are sufficient for acceptance for publication in Cancers.